# Risk of Heart Disease after Cholecystectomy: A Nationwide Population-Based Cohort Study in South Korea

**DOI:** 10.3390/jcm10153253

**Published:** 2021-07-23

**Authors:** Yoo Jin Kim, Young Soo Park, Cheol Min Shin, Kyungdo Han, Sang Hyun Park, Hyuk Yoon, Nayoung Kim, Dong Ho Lee

**Affiliations:** 1Department of Internal Medicine, Seoul National University Bundang Hospital, Seongnam 13620, Korea; eugenek518@gmail.com (Y.J.K.); dkree@naver.com (Y.S.P.); bodnsoul@hanmail.net (H.Y.); nayoungkim49@empas.com (N.K.); dhljohn@yahoo.co.kr (D.H.L.); 2Department of Statistics and Actuarial Science, Soongsil University, Seoul 06978, Korea; 3Department of Medical Statistics, College of Medicine, Catholic University of Korea, Seoul 06591, Korea; ujk8774@naver.com

**Keywords:** cholecystectomy, congestive heart failure, myocardial infarction, atrial fibrillation

## Abstract

The aim of the study is to evaluate the risk of heart disease in individuals who underwent cholecystectomy. This was a retrospective cohort study using the National Health Insurance Service database of South Korea. A total of 146,928 patients who underwent cholecystectomy and 268,502 age- and sex-matched controls were compared. Multivariate Cox proportional hazard regression analysis was used to estimate the hazard ratio (HR) and 95% confidence interval (CI) for heart disease after cholecystectomy. In results, a previous history of cholecystectomy increased the risk of heart disease (congestive heart failure [CHF], myocardial infarction [MI], atrial fibrillation [AF]) (adjusted HR [aHR]: 1.40, 95% CI: [1.36–1.44]). The increased risk was particularly seen for CHF (1.22 [1.16–1.29]) but not for MI and AF (*p* > 0.05). In the subgroup analyses, cholecystectomy was associated with an increased risk of MI in patients aged <65 years (1.49 [1.16–1.92] and 1.18 [1.05–1.35] in patients aged 40–49 and 50–64 years, respectively), but not in those aged ≥ 65 years (0.932 [0.838–1.037]). Moreover, the risk of MI was increased in patients without diabetes mellitus (DM) (1.16 [1.06–1.27]); however, it was decreased in patients with DM (0.83 [0.72–0.97]). In contrast, cholecystectomy did not modify the risk of AF in the subgroup analyses (all *p* > 0.05). In conclusion, a history of cholecystectomy is associated with an increased risk of CHF. Cholecystectomy may increase the risk of MI in the younger population without DM. These findings suggest that the alteration of bile metabolism and homeostasis might be potentially associated with the development of some heart diseases.

## 1. Introduction

Cholecystectomy is the gold standard for symptomatic gallstone disease, acalculous cholecystitis, biliary dyskinesia, gallbladder polyps, and gallstone pancreatitis [1]. Laparoscopic cholecystectomy (LC), which is less invasive and more cost-effective [2,3,4], is now a preferred therapeutic option [5]. Cholecystectomy is the fifth most common surgical procedure in Korea with an annual incidence of over 70,000 per year [6]. Despite an increasing number of patients undergoing cholecystectomy, the long-term effects of cholecystectomy have not been fully elucidated. However, emerging evidence has indicated that cholecystectomy could result in an increased risk of metabolic diseases, such as dyslipidemia and non-alcoholic fatty liver disease (NAFLD) [7,8,9]. Heart diseases are known to be one of the leading causes of death, regardless of sex and race, in the United States [10]. At the same time, the socioeconomic burden of heart disease is now considerably significant and rising steadily and efforts have been to modify and identify risk factors for prevention of heart disease globally [11,12,13,14]. Although coronary artery disease (CAD) and gallstone disease share combined risk factors such as metabolic syndrome [15], the potential association between cholecystectomy and CAD is questionable. Some studies claim that cholecystectomy may be associated with an increased incidence of CAD [16,17], but this association remains controversial [18].

Cholecystectomy is a frequently performed procedure globally [19]. Laparoscopic cholecystectomy is considered as a gold standard for the treatment of gallstone disease [20,21]. Meanwhile, the incidence and disease burden of heart diseases are consistently increasing. However, there is a lack of studies examining the relationship between cholecystectomy and heart diseases. Thus, we performed a nationwide cohort study to elucidate the risk of heart diseases, including congestive heart failure (CHF), atrial fibrillation (AF), and myocardial infarction (MI) in patients who underwent cholecystectomy compared to the general population.

## 2. Materials and Methods

### 2.1. Data Source

This was a retrospective cohort study using the database of the National Health Insurance Service (NHIS) of South Korea. Almost 97% of Koreans are under the mandatory health insurance system. The NHIS database contains data on basic information of the enrollees, such as age, sex, income, and residency, as well as records of healthcare visits, admission events including medical procedures, prescriptions, incurred expenses, and diagnoses based on the International Classification of Diseases, 10th revision (ICD-10). The NHIS has additional data on the health check-ups of all subscribers who are over 40 years of age or employees of any age. This is because they are offered a biennial national health check-up program, which includes the results of laboratory tests, as well as their blood pressure, body mass index, smoking status, alcohol consumption, and exercise levels. More details about the NHIS in South Korea have been reported previously [22].

Informed consents from the enrollees were not required because the study was based on routinely collected medical data. The study was approved by the Institutional Review Board of the Seoul National University Bundang Hospital (IRB number: X-2003/601-904).

### 2.2. Study Population and Design

We initially recruited 339,870 NHIS enrollees who underwent cholecystectomy between 1 January 2010 and 31 December 2015. For the control group, 679,740 subjects were initially included in a ratio of 1:2 in the age- and sex-matched groups compared to the cholecystectomy group. For the participants who had undergone cholecystectomy, their index dates were defined as the date of cholecystectomy. For controls, it was assigned to each control as the index date of the matched case. In cholecystectomy group, 192,042 patients were excluded (lack of two years of medical examination record prior to index date, 147,686 patients; missing variable, 5865 patients; age under 40, 21,999 patients; previous history of heart disease before enrollment, 12,794; heart disease developed within 1 year following the enrollment, 3698). In the control group, 411,238 subjects were excluded using the same criteria as those in the cholecystectomy group (lack of two years of medical examination record prior to index date, 342,242 subjects; missing variable, 8666 subjects; age under 40, 42,337 subjects; previous history of heart disease before enrollment, 14,583; heart disease developed within 1 year following the enrollment, 341). Finally, a total of 146,928 patients who underwent cholecystectomy and 268,502 subjects who did not undergo cholecystectomy were compared in the final analysis (Figure 1). We identified cholecystectomy operations using the corresponding insurance claim codes Q7380. The subjects in the control group were from the general population and had no history of cholecystectomy or previous heart disease; MI (ICD-10 codes I21-24), CHF (ICD-10 code I50), and AF (ICD-10 code I48). The control group subjects were evaluated simultaneously with the cholecystectomy group patients from the index date.

The primary outcome was the incidence of heart disease (MI, CHF, and AF). Information on comorbidities (including diabetes mellitus, hypertension, and dyslipidemia), date of diagnosis of MI/CHF/AF, and latency period were extracted from the NHIS database.

### 2.3. Covariates

Basic information from the NHIS database was used to collect data for variables that were regarded as risk factors for heart disease and for covariates in multivariable analyses, including age, sex, income (lowest 20% of the income bracket, who receive Medicaid program), residence (metropolitan, city, and rural), alcohol intake (none, heavy drinker), cigarette smoking (never, former, and current), body mass index (BMI), and regular exercise. Subjects were classified into three age groups: relatively younger subjects between 40 and 49 years, middle-aged subjects between 50 and 64 years, and older subjects > 65 years. Hypertension, diabetes mellitus, and dyslipidemia were defined using ICD-10 codes—I10–I15, E11–E14, and E78, respectively and noted based on relevant prescriptions. We then analyzed the characteristics of subjects stratified according to their age groups as well as the presence of comorbidities.

### 2.4. Statistical Analyses

The chi-square test (or Fisher exact test when any expected cell count was less than 5 for a 2 × 2 table) was used to compare the categorical variables between the cholecystectomy and control groups. The incidence of heart disease between the two groups was calculated per 1000 person-years, and multivariate Cox proportional hazard regression analysis was used to estimate the hazard ratio (HR) and 95% confidence interval (CI) for heart disease after cholecystectomy. HR was calculated by comparing the ratio of heart disease in the cholecystectomy group to that in the control group, as well as adjusting for age, sex, household income level, place of residence, diabetes mellitus (DM), hypertension, dyslipidemia, smoking status, exercise regularity, alcohol consumption, and BMI. Statistical analyses were performed using the SAS version 9.4 software package (SAS Institute, Cary, NC, USA), and results with two-sided *p*-values of <0.05 were considered significant.

## 3. Results

### 3.1. Characteristics of the Study Population

As mentioned above, a total of 146,928 patients underwent cholecystectomy and 268,502 patients did not undergo cholecystectomy (Figure 1). The characteristics of the study population are summarized in Table 1.

Of the 146,928 patients who underwent cholecystectomy, 74,622 (50.79%) were males, and the mean age in the cholecystectomy group was 58.35 ± 11.01 years. In the control group, 136,271 (50.75%) patients were male, and the mean age was 57.68 ± 11.01 years, with most patients between 50 and 64 years of age. The prevalence of hypertension, DM, and dyslipidemia was higher in the cholecystectomy group than in the control group (41.61% vs. 38.23%, 18.13% vs. 14.11%, and 30.6 4% vs. 27.99%, respectively; all *p* < 0.0001). The control group had a higher rate of non-smokers than that in the cholecystectomy group (62.7% vs. 64.1%, *p* < 0.0001). After a 1-year lag, median follow-up duration was 2.56 years (interquartile range, 1.21–4.00 years).

### 3.2. Incidence and Risk of Heart Diseases in the Cholecystectomy Group Relative to the Matched Control Group

The incidence of heart diseases (AF, MI, and CHF) in the cholecystectomy group was 22.75/1000 person-years while that in the control group was 16.86/1000 person-years. The adjusted HR (aHR) of previous cholecystectomy for heart diseases was 1.40 (95% CI 1.36–1.44, *p* < 0.0001) (Table 2). In the subgroup analysis, the risk of heart disease was more significant in the absence of hypertension, DM, and dyslipidemia (Table 3).

A higher risk of CHF (aHR 1.22, 95% CI 1.16–1.29, *p* < 0.0001) was observed in the cholecystectomy group (Table 2). When stratified by age, the aHRs for CHF in the cholecystectomy group were 1.71 (95% CI 1.39–2.09, *p* < 0.0001), 1.39 (95% CI 1.26–1.54, *p* < 0.0001), and 1.13 (95% CI 1.06–1.21, *p* = 0.0001) for the age groups 40–49, 50–64, and >65 years, respectively (Table 4). Moreover, a higher risk of CHF was observed in the cholecystectomy group without DM (aHR 1.32, 95% CI 1.24–1.40, *p* < 0.0001) than in those with DM (aHR 1.01, 95% CI 0.92–1.12) (Table 5).

There was no significant difference in the risk of MI between the cholecystectomy and control groups. However, in the analysis of age stratification, the aHR for MI in the cholecystectomy group was 1.49 (95% CI 1.16–1.92, *p* = 0.0021), 1.19 (95% CI 1.05–1.35, *p* = 0.0078), and 0.93 (95% CI 0.84–1.04, *p* = 0.1975) in the age groups 40–49, 50–64, and >65 years, respectively (Table 4). Furthermore, the cholecystectomy group without DM had a higher risk of MI (aHR 1.16, 95% CI 1.06–1.27, *p* = 0.0012) than those with DM (aHR 0.83, 95% CI 0.72–0.97, *p* = 0.0171). At the same time, the cholecystectomy group without dyslipidemia also had a higher risk of MI (aHR 1.14, 95% CI 1.03–1.25, *p* = 0.0101) than those with dyslipidemia (Table 5). On the other hand, AF showed no significant difference in the overall or subgroup analysis.

## 4. Discussion

The results of our study showed that the incidence of heart diseases was higher in the cholecystectomy group than in the control group (aHR = 1.40, Table 2). When we classified heart diseases as AF, MI, and CHF, the risk of CHF was higher in the cholecystectomy group (aHR = 1.22), but the risk of AF was not different between the two groups (Table 2). MI also showed no significant association when analyzed in the overall population, but the risk was significantly higher in the cholecystectomy group among patients between 40 and 49 years of age (aHR = 1.49, Table 4). Interestingly, cholecystectomy was negatively associated with the risk of MI among individuals without DM (Table 4).

Despite the advances in the treatment of various heart diseases, their social burden remains high [14,23]. Therefore, identifying high-risk groups may allow for tailored treatment. To the best of our knowledge, this is the first study to report an association between cholecystectomy and the development of various heart diseases. Although we could not verify the underlying mechanism of the association between cholecystectomy and heart diseases in the current study, there are several possible explanations for this association. One of them is that perioperative interruption of antithrombotic agents for cholecystectomy might increase the risk of heart diseases in the high-risk population [24,25]. In this study, however, those who developed heart disease within 1 year after cholecystectomy were excluded.

The main molecules that regulate bile acid metabolism are fibroblast growth factor 19 (FGF19) and nuclear farnesoid X receptor (FXR) [26]. Intestinal bile acid acts as a positive signaling pathway for FXR, which induces FGF19. This ileum-derived FGF19 is transported to the liver and ultimately suppresses bile acid synthesis [26,27,28]. A growing body of evidence recognizes bile acid and FGF19 as metabolic regulators that are involved in the control of energy homeostasis as well as lipid and glucose metabolism [29,30,31,32,33]. While FGF19 is expressed in various human organs, including gallbladder (GB) cholangiocytes [34], cholecystectomy may have varied effects on metabolism. In this regard, Barrera et al. reported that human GB cholangiocytes express and secrete high levels of FGF19 compared to the distal ileum [35]. Plasma bile acid synthesis is reported to increase two-fold after cholecystectomy, and the alteration of diurnal rhythm with a reduced FGF19 noon peak was also observed [35]. Through this study, we can infer a potential link between FGF19 and bile acid metabolism derived from GB cholangiocytes, which can lead to impaired metabolic regulation after cholecystectomy.

Moreover, previous studies have reported that bile acid and the heart might be linked through bile acid receptors in the cardiomyocytes [36]. Several in vitro studies have demonstrated that these receptors play a role in apoptosis or hypertrophy of the cardiomyocytes. A recent study by Pu et al. demonstrated FXR expression in adult cardiomyocytes and cardiac tissue induced by chenodeoxycholic acid (CDCA) [36]. Exposure to CDCA induced time- and dose-dependent apoptosis of FXR-expressing cardiac tissue [36,37]. The Takeda G protein-coupled receptor 5 (TGR5) is another bile acid receptor expressed in the heart but its function is poorly understood [36]. There is an in vivo study which found TGR5 interaction with cardiomyocytes in mice. They administered potent natural TGR5 agonists, namely taurochenodeoxycholic acid and lithocholic acid. This induced up-regulation of protein kinase B, which is known to be related to cardiac hypertrophy [38]. More data are required to obtain further insight regarding the association between bile acids and heart diseases. Nonetheless, these studies indicate that changes in bile acid metabolism and homeostasis can result in heart failure.

There is increasing evidence that changes in bile acid metabolism associated with cholecystectomy can also affect the intestinal microbiome [39,40]. Wang et al. compared the fecal microbiota in a cholecystectomy group and the healthy population. Alteration and loss of diversity of the intestinal microbiota was seen in the cholecystectomy group [36]. Recent studies on the intestinal microbiome have demonstrated some common features of HF and CAD, in particular, a decrease in intestinal microbes with the ability to produce butyrate as well as an increase in circulating levels of trimethylamine-N-oxide [41,42,43]. This series of studies suggests that microbiome dysbiosis after cholecystectomy might be associated with heart disease.

Another possible mechanism for heart disease may be due to the effect of metabolic dysregulation after cholecystectomy. Gallstone disease is the most common cause of cholecystectomy and accounts for 80% of all cholecystectomies. Since 80% of gallstones are cholesterol stones, we can estimate that a considerable portion of cholecystectomies are related to cholesterol stones [44]. This suggests that patients who underwent cholecystectomy have combined risk factors, such as obesity, dyslipidemia, DM, and hypertension. Thus, it has been challenging to evaluate cholecystectomy as an independent risk factor for CAD in previous studies. However, several observational studies have demonstrated an association between cholecystectomy and CAD [16,45]. The gallbladder plays an essential role in the entero-hepatic circulation and homeostasis of bile acids and is regulated by complex neuro-hormonal interactions involving the liver and gut [46]. Gallbladder contraction and refilling is known to be regulated by cholecystokinin, TGR5, and intraluminal bile acids [47]. In addition, the interactions between FGF19 and hepatocyte nuclear FXR are involved in glucose and lipid metabolism [48]. An in vivo study suggested that cholecystectomy itself can cause metabolic deterioration by demonstrating that mice that underwent cholecystectomy had elevated serum and hepatic triglyceride levels and very low-density lipoprotein production [49]. However, in our study, it should be noted that the risk of heart disease was higher in the cholecystectomy group without hypertension, DM, or hyperlipidemia, and a higher risk of MI was observed in the cholecystectomy group without DM. The risk was lower in the cholecystectomy group with DM than in the control group. The mechanisms underlying these results are unclear and should be investigated in further studies.

This study has several limitations. First, it is unclear whether the findings of the study are a causal relationship or simply an association. Second, the definition of heart disease outcomes and comorbidities was based on health insurance claims data. Thus, there is a possibility of misdiagnosis given that the causality of each outcome remains unclear; for example, MI could also be a cause of AF or CHF and vice versa. However, to minimize this possibility, we used an operational definition with a combination of disease codes and hospitalization. Third, there were differences in baseline characteristics between the cholecystectomy and control groups. For this reason, we performed a multivariable analysis of the covariates, as well as stratified analysis of age, hypertension, DM, and dyslipidemia. Fourth, clinical setting (elective vs. emergency cholecystectomy) or surgical methods (open vs. laparoscopic cholecystectomy) or the presence or absence of the complications may have affected patient’s health after cholecystectomy [50], but information on the indication of cholecystectomy, surgical methods, and postoperative complications was lacking in the health insurance claims database. To minimize this, a 1-year lag period was set. Surgery-related factors may have influenced short-term outcomes. Furthermore, it is difficult to explain the reason why the risk of MI after cholecystectomy differs depending on the age group or the presence or absence of diabetes based on these surgical factors alone. Therefore, even considering these limitations, the findings of our study are of clinical importance.

In addition, our study has the advantage of a large sample size using a nationwide cohort that analyzed a diverse array of information. Therefore, we could have access to the records of many patients who underwent cholecystectomy and follow-up periods adequate for investigating the discrepant prognosis of heart disease.

## 5. Conclusions

In this nationwide, retrospective cohort study, we found that the incidence of heart disease was higher in the cholecystectomy group than in the control group. The risk of CHF is higher in the cholecystectomy group regardless of age, but the association may be prominent in younger patients. In addition, the risk of MI may be higher in the cholecystectomy group with patients aged 40–64 years, but not in the older age group. In particular, a higher risk of MI was observed in patients without DM and dyslipidemia. Considering that DM and dyslipidemia play a significant role in the pathogenesis of heart disease, this result implies that unknown mechanisms may be involved in causing physiological changes after cholecystectomy, which need to be clarified in further studies.

## Figures and Tables

**Figure 1 jcm-10-03253-f001:**
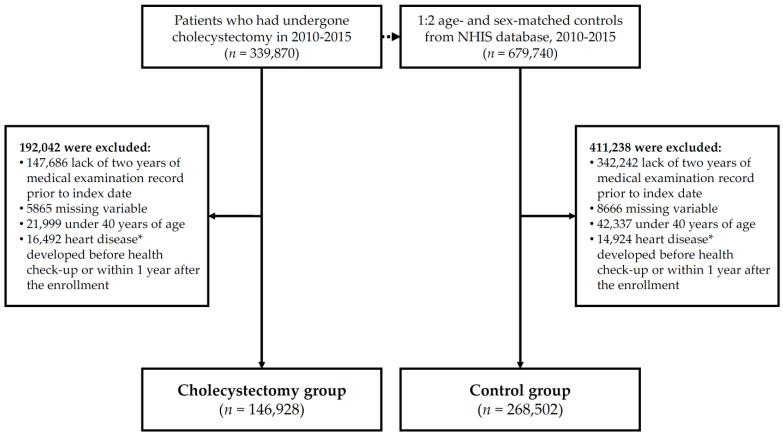
Flow chart of study population. NHIS, National Health Insurance System. * Heart disease includes myocardial infarction, congestive heart failure, and atrial fibrillation.

**Table 1 jcm-10-03253-t001:** Baseline characteristics of the matched cohort.

	Cholecystectomy Cohort (*n* = 146,928)	Matched Control Subjects (*n* = 268,502)	*p*-Value
Age, year			<0.0001
40–49	40,327 (27.45)	66,433 (24.74)	
50–64	65,285 (44.43)	119,351 (44.45)	
≥65	41,316 (28.12)	82,718 (30.81)	
Age (mean ± SD)	57.68 ± 11.01	58.35 ± 11.01	<0.0001
Sex (male)	74,622 (50.79)	136,271 (50.75)	0.8253
Smoking			<0.0001
Never-smoker	92,121 (62.70)	172,112 (64.10)	
Ex-smoker	26,799 (18.24)	47,418 (17.66)	
Current smoker	28,008 (19.06)	48,972 (18.24)	
Comorbidity			
Hypertension	61,139 (41.61)	102,656 (38.23)	<0.0001
Diabetes mellitus	26,638 (18.13)	37,883 (14.11)	<0.0001
Dyslipidemia	45,018 (30.64)	75,148 (27.99)	<0.0001
Chronic kidney disease	12,792 (8.71)	22,837 (8.51)	0.027
Alcohol consumption			<0.0001
None	92,672 (63.07)	163,640 (60.95)	
Mild-moderate drinker	44,750 (30.46)	87,383 (32.54)	
Heavy drinker	9506 (6.47)	17,479 (6.51)	
BMI			<0.0001
<18.5	2808 (1.91)	6985 (2.60)	
18.5–23	43,237 (29.43)	98,580 (36.71)	
23–25	38,278 (26.05)	71,505 (26.63)	
25–30	54,473 (37.07)	82,624 (30.77)	
≥30	8132 (5.53)	8808 (3.28)	
Regular exercise	29,677 (20.20)	57,927 (21.57)	<0.0001
Income, low 20%	30,052 (20.45)	56,319 (20.98)	<0.0001
Place of residence			<0.0001
Metropolitan	88,946 (60.54)	162,406 (60.49)	
City	41,242 (28.07)	73,984 (27.55)	
Rural	16,740 (11.39)	32,112 (11.96)	
BMI	24.62 ± 3.23	23.95 ± 3.06	<0.0001
Fasting blood glucose	103.32 ± 26.69	101.32 ± 24.79	<0.0001
SBP	124.68 ± 14.99	124.71 ± 15.30	0.5197
DBP	77.04 ± 9.83	76.94 ± 9.94	0.0021
Total cholesterol	195.92 ± 38.15	197.58 ± 37.37	<0.0001
HDL cholesterol	52.54 ± 14.63	54.46 ± 14.76	<0.0001
LDL cholesterol	116.75 ± 38.69	116.89 ± 37.63	0.2509

Data are expressed as mean ± SD or *n* (%). *p*-values were calculated using χ^2^ test or Student’s *t*-test. BMI, body mass index; SBP, systolic blood pressure; DBP, diastolic blood pressure; HDL, high-density lipoprotein; LDL, low-density lipoprotein.

**Table 2 jcm-10-03253-t002:** Baseline characteristics of the matched cohort.

Study Outcome		Population (*n*)	Follow-Up Duration (Person-Year)	Events (*n*)	Incidence Rate (per 1000 p-y)	Multivariate 1 HR (95% CI)	Multivariate 2 HR (95% CI)	Multivariate 3 HR (95% CI)
Heart disease *	Control	268,502	716,860.04	12,087	16.861	1 (Ref.)	1 (Ref.)	1 (Ref.)
	Cholecystectomy	146,928	387,925.32	8827	22.7544	**1.350 (1.313, 1.387)**	**1.417 (1.379, 1.456)**	**1.399 (1.361, 1.438)**
Myocardial infarction	Control	268,502	716,860.04	1812	2.52769	1 (Ref.)	1 (Ref.)	1 (Ref.)
	Cholecystectomy	146,928	387,925.32	1035	2.66804	1.056 (0.978, 1.140)	1.100 (1.019, 1.187)	1.059 (0.981, 1.144)
Atrial fibrillation	Control	268,502	716,860.04	3045	4.24769	1 (Ref.)	1 (Ref.)	1 (Ref.)
	Cholecystectomy	146,928	387,925.32	1592	4.10388	0.966 (0.910, 1.027)	1.008 (0.949, 1.071)	0.976 (0.918, 1.038)
Congestive heart failure	Control	268,502	716,860.04	3685	5.14047	1 (Ref.)	1 (Ref.)	1 (Ref.)
	Cholecystectomy	146,928	387,925.32	2392	6.16614	**1.200 (1.140, 1.264)**	**1.263 (1.200, 1.330)**	**1.220 (1.158, 1.285)**

Incidence rate: per 1000 person-years. Multivariate model 1: not adjusted. Multivariate model 2: age, sex-adjusted. Multivariate model 3: age, sex, income, place of residence, DM, hypertension, dyslipidemia, smoking, drinking, regular exercise, BMI. * Heart disease includes myocardial infarction, congestive heart failure, and atrial fibrillation. Bold style indicates statistical significance. BMI, body mass index; DM, diabetes mellitus; p-y, person-year; HR, hazard ratio; CI, confidence interval.

**Table 3 jcm-10-03253-t003:** Comorbidity and adjusted hazard ratio of heart disease *.

Comorbidity		Population (*n*)	Follow-Up Duration (Person-Year)	Events (*n*)	Incidence Rate (per 1000 p-y)	Multivariate 1 HR (95% CI)	Multivariate 2 HR (95% CI)	Multivariate 3 HR (95% CI)
Hypertension								
No	Control	165,846	444,968.2	5167	11.6121	1 (Ref.)	1 (Ref.)	1 (Ref.)
	Cholecystectomy	85,789	228,935.8	3779	16.5068	**1.421 (1.363, 1.482)**	**1.527 (1.464, 1.592)**	**1.528 (1.465, 1.594)**
Yes	Control	102,656	271,891.9	6920	25.4513	1 (Ref.)	1 (Ref.)	1 (Ref.)
	Cholecystectomy	61,139	158,989.5	5048	31.7505	**1.248 (1.204, 1.295)**	**1.321 (1.274, 1.370)**	**1.312 (1.265, 1.361)**
Diabetes mellitus								
No	Control	230,619	619,962.1	9164	14.7815	1 (Ref.)	1 (Ref.)	1 (Ref.)
	Cholecystectomy	120,290	320,782.5	6390	19.92	**1.348 (1.305, 1.391)**	**1.453 (1.407, 1.500)**	**1.458 (1.412, 1.506)**
Yes	Control	37,883	96,897.96	2923	30.1658	1 (Ref.)	1 (Ref.)	1 (Ref.)
	Cholecystectomy	26,638	67,142.86	2437	36.2957	**1.204 (1.141, 1.270)**	**1.228 (1.164, 1.296)**	**1.247 (1.182, 1.316)**
Dyslipidemia								
No	Control	193,354	525,170.4	8314	15.8311	1 (Ref.)	1 (Ref.)	1 (Ref.)
	Cholecystectomy	101,910	274,620.1	6121	22.289	**1.408 (1.362, 1.455)**	**1.483 (1.435, 1.533)**	**1.471 (1.423, 1.521)**
Yes	Control	75,148	191,689.7	3773	19.6829	1 (Ref.)	1 (Ref.)	1 (Ref.)
	Cholecystectomy	45,018	113,305.2	2706	23.8824	**1.213 (1.155, 1.275)**	**1.275 (1.213, 1.339)**	**1.251 (1.191, 1.315)**

Incidence rate: per 1000 person-years. Multivariate model 1: not adjusted. Multivariate model 2: age, sex-adjusted. Multivariate model 3: Age, sex, income, place of residence, DM, hypertension, dyslipidemia, smoking, drinking, regular exercise, BMI. * Heart disease includes myocardial infarction, congestive heart failure, and atrial fibrillation. Bold style indicates statistical significance. BMI, body mass index; DM, diabetes mellitus; p-y, person-year; HR, hazard ratio; CI, confidence interval.

**Table 4 jcm-10-03253-t004:** Incidence of heart disease in the cholecystectomy and control groups according to age-group classification.

Age, Year		Population (*n*)	Follow-Up Duration (Person-Year)	Events (*n*)	Incidence Rate (per 1000 p-y)	Multivariate 1 HR (95% CI)	Multivariate 2 HR (95% CI)	Multivariate 3 HR (95% CI)
Myocardial infarction								
Total	Control	268,502	716,860.04	1812	2.52769	1 (Ref.)	1 (Ref.)	1 (Ref.)
	Cholecystectomy	146,928	387,925.32	1035	2.66804	1.056 (0.978, 1.14)	**1.100 (1.019, 1.187)**	1.059 (0.981, 1.144)
40–49	Control	66,433	179,569.62	124	0.69054	1 (Ref.)	1 (Ref.)	1 (Ref.)
	Cholecystectomy	40,327	108,316.12	121	1.1171	**1.617 (1.259, 2.078)**	**1.616 (1.258, 2.076)**	**1.488 (1.155, 1.917)**
50–64	Control	119,351	325,231.05	603	1.85407	1 (Ref.)	1 (Ref.)	1 (Ref.)
	Cholecystectomy	65,285	175,332.72	408	2.327	**1.255 (1.107, 1.423)**	**1.257 (1.108, 1.425)**	**1.187 (1.046, 1.347)**
≥65	Control	82,718	212,059.37	1085	5.11649	1 (Ref.)	1 (Ref.)	1 (Ref.)
	Cholecystectomy	41,316	104,276.48	506	4.85248	0.949 (0.854, 1.055)	0.949 (0.854, 1.054)	0.932 (0.838, 1.037)
Atrial fibrillation								
Total	Control	268,502	716,860.04	3045	4.24769	1 (Ref.)	1 (Ref.)	1 (Ref.)
	Cholecystectomy	146,928	387,925.32	1592	4.10388	0.966 (0.910, 1.027)	1.008 (0.949, 1.071)	0.976 (0.918, 1.038)
40–49	Control	66,433	179,569.62	207	1.15276	1 (Ref.)	1 (Ref.)	1 (Ref.)
	Cholecystectomy	40,327	108,316.12	154	1.42176	**1.233 (1.001, 1.52)**	**1.233 (1.001, 1.519)**	1.172 (0.949, 1.447)
50–64	Control	119,351	325,231.05	980	3.01324	1 (Ref.)	1 (Ref.)	1 (Ref.)
	Cholecystectomy	65,285	175,332.72	603	3.43918	**1.142 (1.032, 1.264)**	**1.143 (1.033, 1.265)**	1.091 (0.986, 1.209)
≥65	Control	82,718	212,059.37	1858	8.7617	1 (Ref.)	1 (Ref.)	1 (Ref.)
	Cholecystectomy	41,316	104,276.48	835	8.00756	0.914 (0.842, 0.992)	0.914 (0.843, 0.992)	0.897 (0.826, 0.974)
Congestive heart failure								
Total	Control	268,502	716,860.04	3685	5.14047	1 (Ref.)	1 (Ref.)	1 (Ref.)
	Cholecystectomy	146,928	387,925.32	2392	6.16614	**1.200 (1.140, 1.264)**	**1.263 (1.200, 1.330)**	**1.220 (1.158, 1.285)**
40–49	Control	66,433	179,569.62	181	1.00797	1 (Ref.)	1 (Ref.)	1 (Ref.)
	Cholecystectomy	40,327	108,316.12	199	1.83721	**1.823 (1.491, 2.230)**	**1.822 (1.490, 2.229)**	**1.708 (1.394, 2.094)**
50–64	Control	119,351	325,231.05	906	2.78571	1 (Ref.)	1 (Ref.)	1 (Ref.)
	Cholecystectomy	65,285	175,332.72	708	4.03804	**1.452 (1.316, 1.602)**	**1.453 (1.317, 1.603)**	**1.393 (1.262, 1.538)**
≥65	Control	82,718	212,059.37	2598	12.2513	1 (Ref.)	1 (Ref.)	1 (Ref.)
	Cholecystectomy	41,316	104,276.48	1485	14.241	**1.163 (1.091, 1.240)**	**1.163 (1.091, 1.240)**	**1.134 (1.064, 1.210)**

Multivariate model 1: not adjusted. Multivariate model 2: sex-adjusted. Multivariate model 3: sex, income, place of residence, DM, hypertension, dyslipidemia, smoking, drinking, regular exercise, BMI. Incidence rate: per 1000 person-years. HR, hazard ratio; CI, confidence interval; BMI, body mass index; DM, diabetes mellitus; p-y, person-year. Bold style indicates statistical significance.

**Table 5 jcm-10-03253-t005:** Incidence of myocardial infarction, atrial fibrillation, and congestive heart disease in the cholecystectomy and control groups according to comorbidities.

Comorbidity		Population (*n*)	Follow-Up Duration (Person-Year)	Events (*n*)	Incidence Rate (per 1000 p-y)	Multivariate 1 HR (95% CI)	Multivariate 2 HR (95% CI)	Multivariate 3 HR (95% CI)
Myocardial infarction								
Hypertension								
No	Control	165,846	444,968.2	801	1.80013	1 (Ref.)	1 (Ref.)	1 (Ref.)
	Cholecystectomy	85,789	228,935.8	455	1.98746	1.104 (0.984, 1.238)	**1.172 (1.044, 1.315)**	**1.138 (1.014, 1.278)**
Yes	Control	102,656	271,891.9	1011	3.71839	1 (Ref.)	1 (Ref.)	1 (Ref.)
	Cholecystectomy	61,139	158,989.5	580	3.64804	0.982 (0.887, 1.088)	1.028 (0.928, 1.138)	1.003 (0.905, 1.111)
Diabetes mellitus								
No	Control	230,619	619,962.1	1332	2.14852	1 (Ref.)	1 (Ref.)	1 (Ref.)
	Cholecystectomy	120,290	320,782.5	763	2.37856	**1.107 (1.013, 1.210)**	**1.180 (1.079, 1.289)**	**1.159 (1.060, 1.268)**
Yes	Control	37,883	96,897.96	480	4.95366	1 (Ref.)	1 (Ref.)	1 (Ref.)
	Cholecystectomy	26,638	67,142.86	272	4.05106	**0.819 (0.705, 0.950)**	**0.831 (0.716, 0.965)**	**0.834 (0.719, 0.968)**
Dyslipidemia								
No	Control	193,354	525,170.4	1138	2.16692	1 (Ref.)	1 (Ref.)	1 (Ref.)
	Cholecystectomy	101,910	274,620.1	666	2.42517	**1.119 (1.017, 1.232)**	**1.172 (1.065, 1.289)**	**1.135 (1.030, 1.249)**
Yes	Control	75,148	191,689.7	674	3.5161	1 (Ref.)	1 (Ref.)	1 (Ref.)
	Cholecystectomy	45,018	113,305.2	369	3.25669	0.926 (0.816, 1.052)	0.961 (0.847, 1.092)	0.94 (0.827, 1.067)
Atrial fibrillation								
Hypertension								
No	Control	165,846	444,968.2	1260	2.83166	1 (Ref.)	1 (Ref.)	1 (Ref.)
	Cholecystectomy	85,789	228,935.8	625	2.73002	0.964 (0.876, 1.061)	1.026 (0.932, 1.129)	1.011 (0.918, 1.114)
Yes	Control	102,656	271,891.9	1785	6.56511	1 (Ref.)	1 (Ref.)	1 (Ref.)
	Cholecystectomy	61,139	158,989.5	967	6.08216	0.927 (0.857, 1.003)	0.971 (0.898, 1.050)	0.955 (0.882, 1.033)
Diabetes mellitus								
No	Control	230,619	619,962.1	2461	3.9696	1 (Ref.)	1 (Ref.)	1 (Ref.)
	Cholecystectomy	120,290	320,782.5	1204	3.75332	0.946 (0.883, 1.014)	1.010 (0.942, 1.082)	0.982 (0.916, 1.052)
Yes	Control	37,883	96,897.96	584	6.02696	1 (Ref.)	1 (Ref.)	1 (Ref.)
	Cholecystectomy	26,638	67,142.86	388	5.77872	0.959 (0.844, 1.091)	0.976 (0.859, 1.110)	0.959 (0.843, 1.091)
Dyslipidemia								
No	Control	193,354	525,170.4	2125	4.04631	1 (Ref.)	1 (Ref.)	1 (Ref.)
	Cholecystectomy	101,910	274,620.1	1067	3.88537	0.961 (0.893, 1.034)	1.006 (0.934, 1.082)	0.974 (0.904, 1.048)
Yes	Control	75,148	191,689.7	920	4.79942	1 (Ref.)	1 (Ref.)	1 (Ref.)
	Cholecystectomy	45,018	113,305.2	525	4.6335	0.965 (0.867, 1.075)	1.009 (0.906, 1.123)	0.982 (0.881, 1.093)
Congestive heart failure								
Hypertension								
No	Control	165,846	444,968.2	1451	3.26091	1 (Ref.)	1 (Ref.)	1 (Ref.)
	Cholecystectomy	85,789	228,935.8	896	3.91376	**1.199 (1.103, 1.303)**	**1.296 (1.192, 1.409)**	**1.282 (1.179, 1.394)**
Yes	Control	102,656	271,891.9	2234	8.2165	1 (Ref.)	1 (Ref.)	1 (Ref.)
	Cholecystectomy	61,139	158,989.5	1496	9.40943	**1.147 (1.075, 1.225)**	**1.216 (1.139, 1.298)**	**1.185 (1.109, 1.266)**
Diabetes mellitus								
No	Control	230,619	619,962.1	2685	4.33091	1 (Ref.)	1 (Ref.)	1 (Ref.)
	Cholecystectomy	120,290	320,782.5	1707	5.32136	**1.229 (1.156, 1.305)**	**1.330 (1.252, 1.413)**	**1.316 (1.238, 1.399)**
Yes	Control	37,883	96,897.96	1000	10.3201	1 (Ref.)	1 (Ref.)	1 (Ref.)
	Cholecystectomy	26,638	67,142.86	685	10.2021	0.990 (0.898, 1.091)	1.008 (0.915, 1.111)	1.012 (0.918, 1.116)
Dyslipidemia								
No	Control	193,354	525,170.4	2378	4.52805	1 (Ref.)	1 (Ref.)	1 (Ref.)
	Cholecystectomy	101,910	274,620.1	1540	5.60775	**1.239 (1.162, 1.321)**	**1.310 (1.229, 1.397)**	**1.282 (1.202, 1.367)**
Yes	Control	75,148	191,689.7	1307	6.81831	1 (Ref.)	1 (Ref.)	1 (Ref.)
	Cholecystectomy	45,018	113,305.2	852	7.51951	**1.103 (1.012, 1.202)**	**1.163 (1.067, 1.268)**	**1.116 (1.023, 1.217)**

Incidence rate: per 1000 person-years, CI; confidence interval. Multivariate model 1: not adjusted. Multivariate model 2: age, sex-adjusted. Multivariate model 3: age, sex, income, place of residence, DM, hypertension, dyslipidemia, smoking, drinking, regular exercise, BMI. Bold style indicates statistical significance. BMI, body mass index; DM, diabetes mellitus; AF, atrial fibrillation; p-y, person-year.

## Data Availability

The data analyzed in this study is subject to the following licenses/restrictions: We used data from the Korean National Health Insurance Corporation (NHIC) database that only authorized people could access. Requests to access these data sets should be directed to K.H., hkd917@naver.com.

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
