# Peer review of "Risk of Heart Disease after Cholecystectomy: A Nationwide Population-Based Cohort Study in South Korea"

_jcm, 2021, doi:10.3390/jcm10153253_

Round 1

Reviewer 1 Report

Well written and well argued article.
Congratulations on the analysis and the amount of material collected.
The topic is very interesting.
I am a surgeon and I have some questions to ask you:
- were all cholecystectomies performed laparoscopically?
- were all elective cholecystectomies?
In the literature, we know that laparoscopic cholecystectomy is the gold standard for gallbladder disease, but it is burdened by some post-operative complications, by a conversion rate that varies according to the intervention. All of these outcomes vary the patient's postoperative history.
The complicated patient fails to resume his therapy in time and complications affect the patient's health.
I would like to see in your paper a small paragraph that can explain the characteristics of cholecystectomy analyzed in your work (you could also add a table).
You can help yourself by citing these works:
- Mantz J, Samama CM, Tubach F, Devereaux PJ, Collet JP, Albaladejo P, Cholley B, Nizard R, Barré J, Piriou V, Poirier N, Mignon A, Schlumberger S, Longrois D, Aubrun F, Farèse ME, Ravaud P, Steg PG, Stratagem Study Group (2011) Impact of preoperative maintenance or interruption of aspirin on thrombotic and bleeding events after elective non-cardiac surgery: the multicentre, randomized, blinded, placebo-controlled, STRATAGEM Trial. 
Br J Anaesth 107:899-910.
- Vaccari S, Lauro A, Cervellera M, Bellini MI, Palazzini G, Cirocchi R, Tonini V, D’Andrea V. Effect of Antithrombotic therapy on postoperative outcome of 538 consecutive emergency laparoscopic cholecystectomies for acute cholecystitis. Two Italian center’s study. Updates in Surgery (2021). ISSN 2038131X; DOI 10.1007/s13304-021-00994-9 Article in Press.
- Wilson SH, Fasseas P, Orford JL, Lennon RJ, Horlocker T, Charnoff NE, Melby S, Berger PB (2003)   Clinical outcome of patients undergoing non-cardiac surgery in the two months following coronary stenting. 
  J Am Coll Cardiol 42:234-240.
- Vaccari S, Cervellera M, Lauro A, Palazzini G, Cirocchi R, Gjata A, Dibra A, Ussia A, Brighi M, Isaj E, Agastra E, Casella G, Di Matteo FM, Santoro A, Falvo L, Tarroni D, D'Andrea V, Tonini V.  Laparoscopic cholecystectomy: which predicting factors of conversion? Two Italian center's studies. Minerva chirurgica (2020) 75 (3), pp. 141-152.
- Gupta V, Jain G.   Safe laparoscopic cholecystectomy: Adoption of universal culture of safety in cholecystectomy. World J Gastrointest Surg. 2019 Feb 27;11:62-84.
- Sain AH.   Laparoscopic cholecystectomy is the current “gold standard” for the treatment of gallstone disease.  Ann Surg. 1996 Nov;224(5):689–90. 

Author Response

1. Were all cholecystectomies performed laparoscopically? Were all elective cholecystectomies? In the literature, we know that laparoscopic cholecystectomy is the gold standard for gallbladder disease, but it is burdened by some post-operative complications, by a conversion rate that varies according to the intervention. All of these outcomes vary the patient's postoperative history. The complicated patient fails to resume his therapy in time and complications affect the patient's health. I would like to see in your paper a small paragraph that can explain the characteristics of cholecystectomy analyzed in your work (you could also add a table).

Answer: Thank you for your accurate comments. We totally agree with the reviewer’s concerns. In this study, cholecystectomy group included those for gallbladder removal surgery (Q7380) only; those who underwent radical cholecystectomy of gallbladder cancer (Q7410) were not included. In contrast, it was not possible to distinguish whether cholecystectomy was performed laparoscopically using the Korean insurance claim database. However, the study participants were enrolled from 2010 to 2015 when most cholecystectomy was performed laparoscopically in Korea.

In this study, 73,412 (21.7 %) of 339,870 patients underwent cholecystectomy within 3 days of admission to the emergency room; they are regarded as receiving “emergency” cholecystectomy. However it is very difficult to determine who had severe acute cholecystitis correctly using the health insurance claim data, because some patients with severe acute cholecystitis receive cholecystectomy within 72 hours (emergency cholecystectomy) but some of them are treated medically with antibiotics, or with percutaneous transhepatic gallbladder drainage or PTGBD, followed by elective cholecystectomy ≥ 6 weeks later.

In the present study, however, since a 1 year lag period was set, heart disease occurring within 1 year after cholecystectomy was excluded. Surgery-related factors may have influenced the short-term outcome. Furthermore, it is difficult to explain why the risk of myocardial infarction after cholecystectomy differs depending on the age group or the presence or absence of underlying diseases such as diabetes, solely due to these surgical factors. Therefore, even considering these limitations, the findings of our study are interesting and need to be confirmed through more studies in the future.

We also agree that the clinical setting (elective vs. emergency cholecystectomy) or surgical methods (open vs. laparoscopic cholecystectomy) or the presence or absence of the complications may affect the patient’s health after cholecystectomy. Reviewer’s concerns were addressed as a limitation as follows (lines 282-291, page 10): “... Fourth, clinical setting (elective vs. emergency cholecystectomy) or surgical methods (open vs. laparoscopic cholecystectomy) or the presence or absence of the complications may have affected patient’s health after cholecystectomy [49], but information on the indication of cholecystectomy, surgical methods, and postoperative complications was lacking in the health insurance claims database. To minimize this, a 1 year lag period was set. Surgery-related factors may have influenced short-term outcomes. Furthermore, it is difficult to explain the reason why the risk of MI after cholecystectomy differs depending on the age group or the presence or absence of diabetes based on these surgical factors alone. Therefore, even considering these limitations, the findings of our study are of clinical importance. …” Please understand our situation that the study was performed using the Korean insurance claim data. Thank you!

2. You can help yourself by citing these works:

1) Mantz J, Samama CM, Tubach F, Devereaux PJ, Collet JP, Albaladejo P, Cholley B, Nizard R, Barré J, Piriou V, Poirier N, Mignon A, Schlumberger S, Longrois D, Aubrun F, Farèse ME, Ravaud P, Steg PG, Stratagem Study Group (2011) Impact of preoperative maintenance or interruption of aspirin on thrombotic and bleeding events after elective non-cardiac surgery: the multicentre, randomized, blinded, placebo-controlled, STRATAGEM Trial. Br J Anaesth 107:899-910.
2) Vaccari S, Lauro A, Cervellera M, Bellini MI, Palazzini G, Cirocchi R, Tonini V, D’Andrea V. Effect of Antithrombotic therapy on postoperative outcome of 538 consecutive emergency laparoscopic cholecystectomies for acute cholecystitis. Two Italian center’s study. Updates in Surgery (2021). ISSN 2038131X; DOI 10.1007/s13304-021-00994-9 Article in Press.
3) Wilson SH, Fasseas P, Orford JL, Lennon RJ, Horlocker T, Charnoff NE, Melby S, Berger PB (2003) Clinical outcome of patients undergoing non-cardiac surgery in the two months following coronary stenting. J Am Coll Cardiol 42:234-240.
4) Vaccari S, Cervellera M, Lauro A, Palazzini G, Cirocchi R, Gjata A, Dibra A, Ussia A, Brighi M, Isaj E, Agastra E, Casella G, Di Matteo FM, Santoro A, Falvo L, Tarroni D, D'Andrea V, Tonini V.  Laparoscopic cholecystectomy: which predicting factors of conversion? Two Italian center's studies. Minerva chirurgica (2020) 75 (3), pp. 141-152.
5) Gupta V, Jain G. Safe laparoscopic cholecystectomy: Adoption of universal culture of safety in cholecystectomy. World J Gastrointest Surg. 2019 Feb 27;11:62-84.
6) Sain AH. Laparoscopic cholecystectomy is the current “gold standard” for the treatment of gallstone disease. Ann Surg. 1996 Nov;224(5):689–90. 

Answer: Thank you for your kind suggestions. I read all the articles, and 4 of them (Ref#20, #23, #24, #49 in the revised manuscript) were cited in the Introduction or Discussion sessions accordingly.

The authors really appreciated the reviewer’s kind and accurate comments. The revision based on these comments made this manuscript more accurate and the quality improved. Thank you again.

Cheol Min Shin, M.D., Ph.D.

Reviewer 2 Report

This is an extremely well written and referenced paper.

It is scientifically well designed and its findings are probably meaningful in the Korean population particularly because of the high number of patients and the quality of the health data.

A number of possible mechanisms for the association between heart disease and cholecystectomy are postulated and well explored in the discussion, but is is still possible that this is an association rather than a cause and effect phenomena.

All pages of the manuscript have been labeled 6 of 12.

Author Response

1. This is an extremely well-written and referenced paper. It is scientifically well designed and its findings are probably meaningful in the Korean population particularly because of the high number of patients and the quality of the health data. A number of possible mechanisms for the association between heart disease and cholecystectomy are postulated and well explored in the discussion, but is still possible that this is an association rather than a cause and effect phenomena.

Answer: Thank you for your important comments. We agree with the reviewer’s concerns. It is unclear whether the findings of this study are causal relationship or simply associations. Therefore, more studies are warranted to clarify this issue. We addressed this as a limitation in the Discussion session (lines 273-274, page 10): “… This study has several limitations. First, it is unclear whether the findings of the study are causal relationship or simply associations. …” Thank you!

2. All pages of the manuscript have been labeled “6 of 12”.

Answer: Thank you for finding the error. Page labels were revised in the manuscript. 

The authors really appreciated the reviewer’s kind and accurate comments. The revision based on these comments made this manuscript more accurate and the quality improved. Thank you again.

Cheol Min Shin, M.D., Ph.D.
